# OpenReview forum: "Deliberative Dynamics and Value Alignment in LLM Debates"
_ICLR.cc/2026/Conference — Submitted to ICLR 2026_

### Official Review · Reviewer_N9qX · 2025-10-25

**Soundness:** 2
**Presentation:** 2
**Contribution:** 2
**Rating:** 4
**Confidence:** 4

**Summary:**

The paper studies how deliberation format shapes value expression and consensus in LLM-LLM debates over everyday moral dilemmas. Using 1,000 AITA cases, the authors run pairwise and three-way debates among GPT-4.1, Claude 3.7 Sonnet, and Gemini 2.0 Flash in two settings: synchronous (parallel) and round-robin (sequential). They quantify model inertia and conformity via a multinomial model, analyze verdict change rates, and classify values in explanations using a pruned set of 48 values drawn from “Values in the Wild” (Anthropic) with a separate judge model. Prompt tweaks that explicitly encourage consensus increase revision but do not dramatically raise consensus rates. The paper argues that sociotechnical alignment depends on interaction protocol, not only on single-turn outputs.

**Strengths:**

- Moves beyond accuracy-centric MAD papers by focusing on moral judgments in realistic, ambiguous dilemmas, aligning with calls to evaluate real-world impact and sociotechnical properties.
- The synchronous vs. round-robin comparison neatly surfaces inertia vs. within-round conformity. The multinomial modeling of α (inertia) and γ (conformity) is a clean abstraction that other groups can reuse.
- Adapting Anthropic’s “Values in the Wild” taxonomy for everyday dilemmas gives a concrete lens on which values drive alignment and which are “inherited” at revision. This bridges debate mechanics with value elicitation work.

**Weaknesses:**

- Value labels come from an LLM judge. This creates a risk that the same families of models define and then satisfy the value metric. There is no human audit of value annotations, no adjudication protocol, and no inter-judge agreement statistics.
- Most experiments appear to be one pass per case per setting. Debate outcomes can be stochastic with temperature 1. Report replicate variance on a random 10-20 percent subset, or do bootstrap resampling at the dialogue level to confirm the stability of α and γ estimates.
- Missing literature: https://arxiv.org/abs/2506.12657
- The work argues that protocol choices shape social outcomes, but it does not measure helpfulness or user-centric welfare in the outputs, only verdict agreement and value similarity. Even a small human rating on perceived fairness, empathy, or harm reduction would make the claim more actionable.

**Questions:**

- How sensitive are the value-similarity trends to the judge choice and temperature? Please report a small ablation with a different judge family and a lower temperature.
- You note that GPT steers toward NTA when first, but loses that effect when Claude is second. Can you quantify this steering as an estimated average treatment effect with bootstrapped CIs across orders?
- The “balanced goals” prompt increases revision but not consensus. Can you show per-value inheritance changes under that prompt, and whether revisions move toward the majority or create swaps?

---

> ### Author Response · Authors · 2025-11-24
> **Response to Reviewer N9qX, Weakness 1**
>
> We thank the reviewer for their thoughtful review, and anticipate their feedback will improve the quality of the paper.
>
> **Weakness 1**
>
> > Value labels come from an LLM judge. This creates a risk that the same families of models define and then satisfy the value metric. There is no human audit of value annotations, no adjudication protocol, and no inter-judge agreement statistics.
>
> > How sensitive are the value-similarity trends to the judge choice and temperature? Please report a small ablation with a different judge family and a lower temperature.
>
> This is a vital point. We report on in-progress results to validate the usage of the judge. We are currently completing the following experiments to validate our LLM judge, and will report once we are complete:
>
> - A set of human annotations on 150 randomly sampled deliberations (stratified across verdict and model)
> - A repeated LLM judge with Gemini 2.5 under the exact configurations of the paper
> - An LLM judge with Gemini 2.5 Flash at temperature 0
> - An LLM judge with GPT-5 (temperature cannot be set to 0 for this model).
>
> ---
>
> *Update, Nov 29: See the comment below for an update on this analysis.*

---

> ### Author Response · Authors · 2025-11-24
> **Response to Reviewer N9qX, Weakness 2**
>
> **Weakness 2**
>
> >Most experiments appear to be one pass per case per setting. Debate outcomes can be stochastic with temperature 1. Report replicate variance on a random 10-20 percent subset, or do bootstrap resampling at the dialogue level to confirm the stability of α and γ estimates.
>
> We calculated model fits using bootstrap resampling at the dilemma/deliberation level. In Table 1, we report the confidence intervals, and find that the model estimates of  $\alpha$ and $\gamma$ to be very stable, with the exception of Gemini’s conformity factor for previous rounds (which exhibits no significance).

---

> ### Author Response · Authors · 2025-11-24
> **Response to Reviewer N9qX, Weakness 3**
>
> **Weakness 3**
> > Missing literature: https://arxiv.org/abs/2506.12657
>
> We thank the reviewer for pointing out this highly relevant, high quality work. We will incorporate it into our Related Works section, with a proper treatment of how our work relates to and builds upon this paper.
>
> ---
>
> *Update, Nov 29: We have updated the Related Works to emphasize this paper.*

---

> ### Author Response · Authors · 2025-11-24
> **Response to Reviewer N9qX, Weakness 4**
>
> **Weakness 4**
> > The work argues that protocol choices shape social outcomes, but it does not measure helpfulness or user-centric welfare in the outputs, only verdict agreement and value similarity. Even a small human rating on perceived fairness, empathy, or harm reduction would make the claim more actionable.
>
> This is a salient point, and related to Reviewer ck6d’s feedback on the practical importance of this work. We hope our response to that reviewer is relevant to this reviewer’s concerns about the actionable nature of our work.
>
> In the current paper, our claims are deliberately scoped to internal deliberation behavior (verdict revision, value similarity, inertia and conformity) and not to a full welfare analysis. That said, we agree it is useful to add at least a small human-centered check. We aim to add a light-weight annotation study on a subset of dilemmas, where annotators will rate whether the explanations after deliberation exhibit increased consideration of both parties’ perspectives (e.g., acknowledging each side’s concerns and constraints) as a precise operationalization of “perceived empathy”.
>
> We are more cautious about directly annotating fairness or harm reduction in this work. These concepts will likely be dominated by the annotators’ own value systems, requiring a more carefully designed normative framework and larger-scale annotation effort than we can add within the review period. We will note this explicitly in the Discussion as proposed future work.
>
> ---
>
> *Update, Dec 2:* We conducted a small annotation based on a strict definition of empathy, i.e., sufficiently considering both parties' perspectives for 30 deliberations. Our analysis returned negative results: we did not find that deliberation meaningfully increased empathy according to this definition because all models exhibited aspects of considering both parties at the very outset of deliberation.
>
> Ultimately, our paper does not necessarily suggest that these protocols should *improve* outcomes; rather, they shape them, and it is important to understand how. We hope our paper brings to light these issues and leave further determination of the best protocol for certain outcomes as future work.

---

> ### Author Response · Authors · 2025-11-24
> **Response to Reviewer N9qX, Question 2**
>
> **Question 2**
> > You note that GPT steers toward NTA when first, but loses that effect when Claude is second. Can you quantify this steering as an estimated average treatment effect with bootstrapped CIs across orders?
>
> We thank the reviewer for this suggestion. Our original wording ("GPT steers…") implies a stronger causal claim than we intended. There does, though, exist an order dependent difference in the NTA rate across verdicts. We calculated this as $\Delta(\text{NTA}) = 0.136$, with a 95% bootstrapped CI of $[0.107, 0.165]$. We believe this calculation is in line with what the reviewer asked (though please correct us if we interpreted your suggestion incorrectly).
>
> This suggests that the NTA rate drops by 13.6% when Claude is second rather than third. This may stem from the fact that Claude is less conforming than Gemini, and further that conformity effects compound when multiple models agree in a row.

---

> ### Author Response · Authors · 2025-11-29
> **Response to Reviewer N9qX, Weakness 1: Update**
>
> We provide an update to our response to Weakness 1 in a separate comment since the update is substantial.
>
> We conducted the following validation checks to assess the LLM judge (Gemini 2.5 Flash) we used to classify values in dilemmas:
>
> - **Human validation.** We conducted a human validation of 100 judgments rendered by the models. We chose these judgments to be stratified across model and verdict, so that all three models were equally represented, and all verdicts were represented with respect to their occurrences. One of the authors carried out the annotations.
> - **Repeated LLM Judge.** We conducted the LLM-as-Judge value classification twice, using Gemini 2.5 Flash, in order to assess consistency between runs.
> - **LLM Judge at Temperature 0.** We repeated the LLM-as-Judge value classification using Gemini 2.5 Flash, but with a temperature of 0.
> - **LLM Judge with Different Model.** We conducted the LLM-as-Judge classification using GPT-5 (at temperature 1, since OpenAI reasoning models cannot use lower temperatures).
>
> Value classification is a task with a high degree of intersubjectivity, where different annotators may reasonably disagree on the values present. Thus, we should not expect perfect consistency on this task, but we do expect higher value similarities on the same judgment compared to value similarities between judgments. We note that value similarities for judgments in disagreement ranged from 0.25 to 0.30, while value similarities for judgments in agreement ranged from 0.40 to 0.48.
>
> We report the average value similarity across dilemmas, between the original Gemini 2.5 Flash classification, and the alternate judges. That is, let $\mathcal{V}_i^0$ be the value set for judgment $i$ from the original judge, and let $\mathcal{V}_i^R$ be the value set obtained from the alternate judge; we report $\frac{1}{N}\sum_i J(\mathcal{V}_i^R, \mathcal{V}_i^0)$ where $i$ iterates over experiments and judgments, and $N$ is the number of judgments considered. For the human validation, this is $N = 100$; otherwise, it is the total number of messages across all deliberations.
>
> Our results are reported in the table below. We find that Gemini 2.5 Flash is very self consistent, with value similarities of roughly 0.64. Meanwhile, the human and GPT-5 value similarities are lower (roughly 0.53), but still above what we find in our results in the main text. To put these numbers in perspective, for two value sets with cardinality of 5, their Jaccard similarity would be 0.42 if they shared 3 values and 0.66 if they shared 4 values. Thus, our robustness checks can roughly be interpreted as Gemini 2.5 Flash generally maintaining consistency on 4 out of 5 values. Meanwhile, the human validation and GPT-5.1 generally have 3 to 4 values in agreement.
>
> | Validation Approach             | Value Similarity |
> |--------------------------------|------------------|
> | Human                          | 0.533            |
> | Repeated Gemini 2.5 Flash      | 0.638            |
> | Gemini 2.5 Flash, $T = 0$      | 0.644            |
> | GPT-5                          | 0.547            |
>
> Lastly, we reproduced Figure 5, but with the value sets provided by the GPT-5 judge. We found the same patterns: value similarities that are significantly higher during consensus versus disagreement. We further found that value similarities increased significantly more when reaching consensus after initial disagreement compared to deliberations never reaching consensus. The percentage increase in value similarities for these scenarios was roughly 2.5–3 times higher for consensus-reaching deliberations vs. no-consensus deliberations. Thus, our results are robust to the choice of judge.
>
> **All of the above findings are provided in Appendix I.1.**

---

### Official Review · Reviewer_JE5M · 2025-10-29

**Soundness:** 3
**Presentation:** 3
**Contribution:** 2
**Rating:** 4
**Confidence:** 4

**Summary:**

This work studies values elicited from multi-agent debate verdicts, arriving to interesting conclusions across multiple deliberating formats and models. Experiments are done on 1000 questions from the AITA reddit community, with debates from models in {GPT-4.1, Claude 3.7 Sonnet, and Gemini 2.0 Flash}. Results cover aspects including consensus-forming, values orientations, effects of deliberation format and effects of system-prompt-steering.

**Strengths:**

The paper studies an important problem: values elicitation of multi-turn, multi-agent debate. The conformity to or emergence of certain sets of values during conversations is indeed a phenomenon expected but not well-studied. The work improves our understanding of this problem with layered conclusions from observations of dynamics to quantified values inclination and effects of deliberation formatting.

The choice of AITA questions as dataset is a worthwhile novelty, as they are straight-forward to answer but varied in underlying values.

Experiments are carried out with satisfying scale and analysis.

**Weaknesses:**

The use of Reddit questions as datasets raises natural questioning on pre-trained bias on the tested models' verdicts. Would the results be different if using exclusive / synthetic questions?

Another field of results I feel lacking is the models' performances on their own. All results of the models' values / verdict inclination are drawn during debate with other models, but we also need the model's original inclination (i.e. verdict and inclination without debate) as baseline to determine effects of debate.

When comparing values' similarity, only the overlapping proportions are considered. But lack of overlapping doesn't necessarily mean lack of alignment between values: some values foundations are close to each other, e.g. "Family bonds and cohesion" and "Parental care". For example, if both value A and value B are important to question M and question N, it could be that GPT uses A for M, B for N while Gemini uses B for M and A for N. Some form of general analysis is needed

Experiments on steering seem a little lacking. We expect to see effects of more variables such as allowed conversation rounds.

**Questions:**

Please see Weaknesses, where each paragraph is a raised question, with decreasing importance.

---

> ### Author Response · Authors · 2025-11-23
> **Response to Reviewer JE5M, Weakness 1**
>
> We thank Reviewer JE5M for their thoughtful and thorough feedback. We address the weaknesses in order.
>
> **Weakness 1**
>
> > The use of Reddit questions as datasets raises natural questioning on pre-trained bias on the tested models' verdicts. Would the results be different if using exclusive / synthetic questions?
>
> We thank the reviewer for raising this salient concern about the choice of data. There are several ways in which our usage of Reddit data should shape outcomes.
>
> First, the model outputs could be biased if the scenarios we use are taken within the training cutoff of the model, and thus we can have no reasonable expectation that they were not used for pretraining (though it is unclear how prevalent this bias is, given that posttraining could have a stronger impact on the model outputs). We largely avoid this issue by using scenarios posted on dates that are beyond the training cutoff for all models.
>
> Second, the models clearly have established a sense of the norms of the subreddit due to their pretraining. Thus, situating them as judges, particularly in the AITA context, likely influences the labels they output (e.g., see [1], which demonstrates increased sycophancy when LLMs are not situated as judges). We address this in the second paragraph of our discussion section.
>
> We acknowledge that our focus on recent AITA posts means we are not evaluating on a synthetic domain – where we might have more control over the values present – this choice is intentional. Our goal is to study deliberation and value negotiation in the kinds of messy, multi-valued, personal dilemmas that users “in the wild” bring to deployed systems. In that sense, the fact that models have picked up subreddit norms is part of the sociotechnical object of study rather than a confound we can or should completely remove.
>
> Concurrent related work (we thank Reviewer 2 for bringing this work to our attention) has approached similar questions but from the opposite direction, using synthetic dilemmas to control value trade-offs. For example, LitmusValues [2] introduces AIRiskDilemmas, a synthetic benchmark where each scenario explicitly pits two values against each other, and infers models’ value priorities from their aggregate choices. Such synthetic setups nicely address the “exclusive dataset” concern the reviewer raises, but they also abstract away from the everyday context we focus on here. Ultimately, we see these approaches as complementary, with our work more emphasizing the deliberative dynamics and value evolution.
>
> To directly answer the Reviewer’s question, we would likely expect some changes in the precise verdicts assigned given alternative framings of the dilemmas (e.g., a non-AITA context, and speaking to our point above) or less strongly adhering to a judge framework (as [1] has demonstrated). We would expect that deliberation format (or generally the agentic interaction framework, in a deployed context) would have a strong impact on decision-making, particularly with regard to conformity and inertia, which is the main point we aim to emphasize in this work, independent of the specific dilemma source. A thorough comparison between “in-the-wild” Reddit dilemmas and synthetic or exclusive benchmarks (e.g., AIRiskDilemmas) is substantial enough that we view it as important follow-up work. **We have added material to the Discussion section to raise this point.**
>
> [1] Cheng et al., “ELEPHANT: Measuring and understanding social sycophancy in LLMs” (2025).
>
> [2] Chiu et al., “Will AI Tell Lies to Save Sick Children? Litmus-Testing AI Values Prioritization with AIRiskDilemmas” (2025).

---

> ### Author Response · Authors · 2025-11-23
> **Response to Reviewer JE5M, Weakness 2**
>
> **Weakness 2**
>
> > Another field of results I feel lacking is the models' performances on their own. All results of the models' values / verdict inclination are drawn during debate with other models, but we also need the model's original inclination (i.e. verdict and inclination without debate) as baseline to determine effects of debate.
>
> We thank the reviewer for this helpful suggestion. **To address this, we ran each model individually (no debate partner) on the full set of 1,000 dilemmas, three independent times per model, using the same AITA task setup.** We then compared (i) the verdict distributions from these single-model runs to the Round 1 verdict distributions in the synchronous experiments, and (ii) the agreement rates between the synchronous Round 1 verdicts and each of the single-model runs. **We report these findings in Appendix C.**
>
> Table 1 shows that, for GPT-4.1 and Claude 3.7 Sonnet, the verdict distributions in the single-model setting are very similar to those in the first round of synchronous deliberation. Meanwhile, Gemini 2.0 Flash exhibits a similar proportion of YTA verdicts across settings; however, it issues substantially more NTA and fewer ESH verdicts than in the individual runs (where it uses ESH much more heavily). This suggests that Gemini is more sensitive to the “debate” system prompt framing than GPT and Claude.
>
> | Experiment           |   NTA |   YTA |   NAH |   ESH |  INFO |
> |----------------------|------:|------:|------:|------:|------:|
> | GPT (vs. Claude)     | 0.788 | 0.038 | 0.083 | 0.088 | 0.003 |
> | GPT (vs. Gemini)     | 0.849 | 0.043 | 0.048 | 0.054 | 0.006 |
> | GPT (run 1)          | 0.771 | 0.057 | 0.081 | 0.089 | 0.002 |
> | GPT (run 2)          | 0.763 | 0.054 | 0.086 | 0.095 | 0.002 |
> | GPT (run 3)          | 0.753 | 0.062 | 0.092 | 0.091 | 0.002 |
> | Claude (vs. GPT)     | 0.556 | 0.114 | 0.115 | 0.214 | 0.001 |
> | Claude (vs. Gemini)  | 0.554 | 0.151 | 0.083 | 0.210 | 0.002 |
> | Claude (run 1)       | 0.513 | 0.149 | 0.150 | 0.186 | 0.002 |
> | Claude (run 2)       | 0.527 | 0.152 | 0.145 | 0.174 | 0.002 |
> | Claude (run 3)       | 0.508 | 0.152 | 0.147 | 0.191 | 0.002 |
> | Gemini (vs. Claude)  | 0.519 | 0.331 | 0.064 | 0.074 | 0.012 |
> | Gemini (vs. GPT)     | 0.509 | 0.352 | 0.060 | 0.063 | 0.016 |
> | Gemini (run 1)       | 0.265 | 0.387 | 0.097 | 0.249 | 0.002 |
> | Gemini (run 2)       | 0.271 | 0.389 | 0.094 | 0.244 | 0.002 |
> | Gemini (run 3)       | 0.263 | 0.397 | 0.095 | 0.243 | 0.002 |
>
> Table 2 reports the self-agreement between the synchronous Round 1 verdicts and each independent single-model run (i.e., the fraction of dilemmas where the verdict matches). For both GPT-4.1 and Claude 3.7 Sonnet, these agreements are notably higher than the between-model agreement rates in Round 1 (Figure 2a), indicating that these models are internally consistent and that their Round 1 behavior in the synchronous setting is closely aligned with their “solo” behavior. For Gemini, the self-agreement is lower, and closer to its agreement with other models. This is consistent with the observation above that Gemini’s verdict distribution shifts more under the debate framing, and may help explain its higher change-of-verdict rates in later rounds.
>
> | Experiment          |   Individual Run 1 |   Individual Run 2 |   Individual Run 3 |
> |---------------------|-------------------:|-------------------:|-------------------:|
> | GPT (vs. Claude)    |              0.857 |              0.855 |              0.849 |
> | GPT (vs. Gemini)    |              0.874 |              0.867 |              0.858 |
> | Claude (vs. GPT)    |              0.771 |              0.777 |              0.773 |
> | Claude (vs. Gemini) |              0.721 |              0.723 |              0.710 |
> | Gemini (vs. Claude) |              0.542 |              0.541 |              0.544 |
> | Gemini (vs. GPT)    |              0.534 |              0.532 |              0.536 |
>
> These baselines support our main interpretation: GPT and Claude have stable, model-specific verdict inclinations that carry over from the single-model setting into Round 1 of synchronous debate. The subsequent changes we report are driven by deliberation dynamics rather than unstable baselines. Gemini, by contrast, is more sensitive to the deliberative framing itself, which aligns with its greater flexibility (higher CoV rates) in our experiments. **We add a brief appendix section reporting these single-model baselines and will reference it in the Results section when discussing initial verdict distributions and revision patterns.**

---

> ### Author Response · Authors · 2025-11-23
> **Response to Reviewer JE5M, Weakness 3**
>
> **Weakness 3**
>
> > When comparing values' similarity, only the overlapping proportions are considered. But lack of overlapping doesn't necessarily mean lack of alignment between values: some values foundations are close to each other, e.g. "Family bonds and cohesion" and "Parental care". For example, if both value A and value B are important to question M and question N, it could be that GPT uses A for M, B for N while Gemini uses B for M and A for N. Some form of general analysis is needed
>
> We thank the reviewer for pointing this out. We agree that this is an important consideration. Ultimately, we’d want to construct something like a value graph, where complex interactions between values can be mapped out to allow for a more robust value alignment analysis (e.g., perhaps like moral elicitation graphs proposed by Klingefjord et al., 2024). A full treatment of this is beyond the scope of the current paper, but we can already perform a more general analysis to test the robustness of our findings.
>
> *Values in the Wild* offers a taxonomy clustering similar values together. The values we used (taken from the second tier) fall into clusters. Thus, we can account for your concern by adjusting the value similarity metric by accounting for when two values, though not exactly matching, fall in the same cluster (and thus would be more similar than not).
>
> **We developed a modified version of the Jaccard similarity.** First, we identify exact value matches between two explanations. Then, we additionally match values that fall in the same Values in the Wild cluster and allow these to contribute a partial similarity weight to the intersection in the numerator. We recompute value similarities under this modified metric and examine whether our results change. We should expect the absolute similarity scores to increase, but it is *a priori* unclear whether the trends we report (e.g., higher similarity during agreement and increasing similarity when consensus forms) would still hold once cluster-level similarity is taken into account.
>
> **We found that, for this modified metric, the trends are unchanged (Appendix I.1: Table 5, 6, and 7).** That is, value similarities in rounds with agreement continue to be significantly higher than those with disagreement. Indeed, the value similarities increase under the modified metric, though only modestly. Similarly, after initial disagreement, value similarities significantly increased when consensus was reached. These calculations are conducted assuming the “partial similarity” is equal to 0.5, but we find similar results even if we increase the partial similarity to 1 (where we treat values in the same cluster as exchangeable).
>
> Notably, under the modified metric, we do find that the increase in value similarity under consensus formation is a bit smaller (GPT vs Claude: 31% -> 27.6%; Claude vs. Gemini: 60.9% -> 51.3%; Gemini vs. GPT: 52.4% -> 43.3%). This decrease also occurs for the scenarios in which consensus is not attained (6.3%-> 1.1%; 13.1% -> 10.6%; 17.9%->14.2%). Thus, it may be a consequence of the general increase in value similarities under the modified metric.

---

> ### Author Response · Authors · 2025-11-23
> **Response to Reviewer JE5M, Weakness 4**
>
> **Weakness 4**
>
> > Experiments on steering seem a little lacking. We expect to see effects of more variables such as allowed conversation rounds.
>
> We thank the reviewer for this feedback. In response, we added an additional steering experiment in which we ablated the explicit deliberation objective from the system prompt (i.e., removing the “correctness vs. consensus” goal statement) and re-ran the synchronous deliberations. We found that the change-of-verdict (CoV) rates and consensus rates were largely unchanged.
>
> We will continue to add additional steering experiments, to assess the degree to which we can change the deliberation dynamics. In particular, we will aim to steer away from consensus. Per reviewer ck6d’s suggestion, we will consider steering toward certain values, as conducted in DailyDilemmas (Chiu et al., 2025).
>
> Regarding the number of allowed rounds, our experiments already indicate that allowing more than 4 rounds rarely changes the outcome: either consensus is reached within the first few rounds, or deliberation stagnates due to inertia. This is why we fixed the cap at 4 in the main experiments. We will make this explicit in the paper and, in future work, plan to explore more aggressive steering setups (e.g., prompts that explicitly discourage consensus or emphasize independence) where varying the round limit might interact more strongly with the deliberation dynamics.
>
> ---
>
> *Update, Nov 28:* We have now conducted a series of steering experiments that we hope address the reviewer's concerns. These include:
> - **An ablation we where omit the goals section of the system prompt: Appendix B, Figure 13.** We found that this ablation largely unchanged change-of-verdict rates.
> - **An experiment where we steered models to be more adversarial to each other: Appendix B, Figure 15 and Table 2.** We found that change-of-verdict rates generally decreased (in the case of GPT-4.1, they went to 0). We found consensus formation significantly decreased in this setting. However, relative differences between models persisted (CoV rates). See below for the no-consensus rates in all settings (higher means models failed to reach a consensus more often).
> | Pairing                           | Original | Balanced | Adversarial |
> |-----------------------------------|----------|----------|-------------|
> | Claude 3.7 Sonnet vs. GPT-4.1     | 0.094    | 0.062    | 0.184       |
> | Claude 3.7 Sonnet vs. Gemini 2.0 Flash | 0.115    | 0.129    | 0.154       |
> | GPT-4.1 vs. Gemini 2.0 Flash      | 0.174    | 0.073    | 0.239       |
> - **An experiment where we steered toward a certain value: Appendix B, Figure 16 and Table 3.** We found that deliberation outcomes were largely unchanged (Figure 16), but that usage of the *Empathy and Understanding* value (the value we steered toward) strongly increased.
> | Model                                    | Original | Steered | % Increase |
> |------------------------------------------|----------|---------|------------|
> | Claude 3.7 Sonnet (vs. GPT-4.1)          | 0.297    | 0.403   | 26.2%      |
> | GPT-4.1 (vs. Claude 3.7 Sonnet)          | 0.262    | 0.492   | 46.7%      |
> | Claude 3.7 Sonnet (vs. Gemini 2.0 Flash) | 0.293    | 0.378   | 22.6%      |
> | Gemini 2.0 Flash (vs. Claude 3.7 Sonnet) | 0.290    | 0.396   | 26.7%      |
> | GPT-4.1 (vs. Gemini 2.0 Flash)           | 0.244    | 0.477   | 48.9%      |
> | Gemini 2.0 Flash (vs. GPT-4.1)           | 0.311   | 0.433   | 28.2%      |
> Together, these results demonstrate that deliberative dynamics and values can both be steered by prompting, though in the case of the dynamics (CoV rates), the differences may not be meaningful across models.

---

### Official Review · Reviewer_ck6d · 2025-10-31

**Soundness:** 2
**Presentation:** 3
**Contribution:** 2
**Rating:** 4
**Confidence:** 4

**Summary:**

This paper collect 1k everyday dilemmas from Reddit's r/AITA community as the basis for simulate LLM debates. They developed two settings for two models as a pair (synchronous setting: each comment its verdict; head-to-head: one by one between two models). They tested three models (GPT-4.1, Claude 3.7 Sonnet, and Gemini 2.0 Flash) for order effects and verdict revision. They show some behavioural differences (e.g. Gemini 2.0 Flash prioirizied more on empathy).

**Strengths:**

S1 Important topic
The value alignment topic is important for socio-tech. alignment. The work (through may not be perfect) will help initiate more discussions on this theme.

S2 Useful data resource
The effort of clustering and cleaning the large corpus of values by Values in the Wild could help the community to continue build upon this line of research.

**Weaknesses:**

[more important] W1 motivation of the experiments and practical importance
- motivation: why we need two settings (synchronous deliberation vs. head-to-head)
- any practical use cases to justify such debates can help solving the socio-tech. alignemnt as authors claimed.
- See some recommendations in w4.
- another recommendation: what about the performances of current reasoning models?

[important] W2 The raw source of values adopted by authors could lead to biased analysis. caution is needed.
- The Values in the Wild is the work from Anthropic researchers using their own tool to analyze the human users and AI. It may only cover what human users perceived and interact with an AI "chatbot assistant", rather than an AI (which could be many roles in the future applications).
- also it is based on one model series (Claude) which could give indirect advantages on evaluating Claude series models.
- I recommend authors to note the reader early on that the values studied and the findings are only for human-favored values (in chatbot setting).

[more important] W3 Comparison of models experiments is hard to extend to more models due to the bad choice of metric
- Authors compared 3 models ( GPT-4.1, Claude 3.7 Sonnet, and Gemini 2.0 Flash) by making pairs of two models (GPT vs Claude, GPT vs Gemini and Claude vs Gemini). The choice of metric (proportion of dilemmas) is bad since it is difficult for readers and researchers to compare three at the time (need to infer in each model pair e.g. line 252- 260).
- Recommend authors to use order-based metric/ rank-based metric (e.g., battles by elo ratings used in ChatbotArena). One recent value-based work (LitmusValues) [2] also adopt similar approach to allow an extensive scale of evaluation to offer more practical importance.

[more important] W4 Not generalizable findings
- followed by W3, authors only compared 3 models. It is insufficient and potentially misleading to compare them as model series difference (e.g. GPT-4.1 directly concludes as GPT, Claude 3.7 Sonnet directly concludes as Claude, and Gemini 2.0 Flash directly concludes as Gemini)
- several studies [2,3,4] already show that different models in the same model family could have different value preferences.
- similarly, the findings (line 367-368) between different model pairs are interesting but they could have confounding factors (e.g. the opponent effect on the model being tested).
- recommended some stat tests to justify whether the findings can still hold when comparing across three (and ideally even more models).
- section 4 (steering system prompt) could have more in-depth analysis and experiment. For instance, try different prompt variations. Some prior work [3] has done some explorations on system prompts steerability on value preferences using model spec. Authors can take references of them to provide more in-depth analysis to benefit the community and provide more actionable advice to justify their study's practical importance [W1]

[1] ChatbotArena https://arxiv.org/abs/2403.04132
[2] LitmusValues https://arxiv.org/abs/2505.14633
[3] DailyDilemmas
[4] Multilingual trolley problem https://arxiv.org/abs/2407.02273
[5] https://arxiv.org/pdf/2402.06782

**Questions:**

Q1 why use Gemini-2.5 flash as judge? (line 197)

Q2 what are the arrows in figure 3

**Details Of Ethics Concerns:**

The dataset is curated from Reddit Am I the Asshole (p.10 ethic statement). Authors mentioned "Our dataset was restricted to post content and associated verdicts." but we do not know how they curate the dataset.

As far as I know, Reddit requires researchers to submit a form to use their API and web scrapping is no longer allowed by Reddit.
See more in https://www.reddit.com/r/research/comments/1ghaxbe/reddit_data_for_academic_use/

---

> ### Author Response · Authors · 2025-11-23
> **Response to Reviewer ck6d, Part 1**
>
> We thank the reviewer for their critical, thorough, and thoughtful review, and expect that their feedback will improve the paper. We additionally thank the reviewer for their appreciation of the topic’s importance.
>
> We first address the ethics concern and two questions, then proceed to the weaknesses.
>
> > The dataset is curated from Reddit Am I the Asshole (p.10 ethic statement). Authors mentioned "Our dataset was restricted to post content and associated verdicts." but we do not know how they curate the dataset.
>
> We thank the reviewer for pointing this out, as we did not provide enough detail. We sourced our data from AcademicTorrents (e.g., following [1]). We isolated the submissions for r/AmITheAsshole, and obtained the submission IDs. To obtain accurate data and apply our own preprocessing pipeline, we queried the Reddit API using the submission IDs to obtain our final dataset.
>
> We have added text to the Methods section with these details.
>
> [1] Cheng et al., REALM: A Dataset of Real-World LLM Use Cases (2025).
>
> > Q1 why use Gemini-2.5 flash as judge? (line 197)
>
> This was reflective of the financial resources available to us (i.e., what credits we had available through Researcher Access Programs). We have added additional robustness checks described in our response to Reviewer N9qX, and will continue to supplement this line of analysis.
>
> > Q2 what are the arrows in figure 3
>
> They are the proportion of dilemmas reaching consensus for that verdict. The proportion of dilemmas not reaching consensus is indicated by the red triangles on the bottom of the subplots.

---

> ### Author Response · Authors · 2025-11-23
> **Response to Reviewer ck6d: Weakness 1**
>
> **Weakness 1**
>
> > motivation: why we need two settings (synchronous deliberation vs. head-to-head). any practical use cases to justify such debates can help solving the socio-tech. alignemnt as authors claimed.
>
> First, we want to clarify whether the reviewer meant "synchronous deliberation vs. **round-robin** deliberation" (not head-to-head). Synchronous and round-robin deliberation are formalizations of commonly used debate structures and agentic workflows. In multi-agent debate, many works adopt a parallel structure, where all agents answer and view each other’s responses. This structure closely reflects our synchronous setting [1, 2, 3]. Other agentic workflows and studies instead organize agents into sequential pipelines, where later agents explicitly condition on earlier outputs, which is naturally captured by our round-robin format (indeed, this workflow is built directly into autogen) [4]. Thus, our breakdown into these formats reflects our attempt to distill how such deliberation formats can shape multi-model influence, revision, and value alignment under controlled conditions that echo how current multi-agent systems are actually orchestrated.
>
> With regard to practical use cases: the use of agentic workflows, which leverage the aforementioned debate structures, is increasingly studied in cases with practical consequence. For example, a couple of relevant cases for personal advice and mental health include:
> - MentalAgora uses multi-agent debating among counselor personas as the first stage in a mental health support pipeline before generating final responses to users [5]
> - Multi-agent guided interview (MAGI) coordinates 4 agents in a sequential interview workflow for psychiatric assessment [6]
>
> Other practical use cases, partially motivated by the paper the reviewer cites, include:
> - LLM mediation in democratic deliberation (though this centers LLMs as moderators) [7]
> - Multi-agent orchestration of autonomous driving systems, which incorporates both synchronous and round-robin format structures [8]
> - Multi-agent assessment of complex medical scenarios [9]
>
> Underlying the above use cases, work on the governance of AI agents argues that increasingly autonomous, task-executing agents raise new questions around control, oversight, and liability [10]. This elevates the importance of understanding how their internal deliberation protocols affect the recommendations and norms they propagate, particularly as they touch upon personal domains. We further anticipate that there will be agentic workflows incorporating LLMs interacting in deliberation structures that we cannot anticipate. Thus, we hope this work will motivate further investigation along those lines.
>
> Of course, we acknowledge that these practical use cases will not be a one-to-one match with the AITA framework we use. However, such everyday dilemmas – with their complexity, extraneous details, first-person narrative, etc. – will surely be present in the aforementioned practical scenarios described above. We intend for our usage of AITA dilemmas to be intentionally evocative in the spirit of *thick descriptions* [11], where values must be invoked and models are intentionally pushed to render judgments in rich situations rather than in abstract, highly stylized vignettes.
>
> **We appreciate the reviewer emphasizing the practical use cases, and will restructure the introduction to more strongly emphasize these to center the need for such sociotechnically oriented investigations.**
>
> [1] Du et al., "Improving Factuality and Reasoning via Multiagent Debate" (2023).
>
> [2] Pitre et al., "CONSENSAGENT: Towards Efficient and Effective Consensus in Multi-Agent LLM Interactions Through Sycophancy Mitigation" (2025).
>
> [3] Khan et al, "Debating with More Persuasive LLMs Leads to More Truthful Answers" (2024).
>
> [4] Liu et al., "Synthetic Socratic Debates: Examining Persona Effects on Moral Decision and Persuasion Dynamics" (2025).
>
> [5] Lee et al., MentalAgora: A Gateway to Advanced Personalized Care…" (2024)
>
> [6] Bi et al., "MAGI: Multi-Agent Guided Interview for Psychiatric Assessment" (2025).
>
> [7] Tessler et al., "AI can help humans find common ground in democratic debates" (2024).
>
> [8] Wu et al., "Multi-Agent Autonomous Driving Systems with LLMs: A survey of Recent Advances" (2025).
>
> [9] Chen et al., "Enhancing diagnostic capability with multi-agents conversational large language models" (2025).
>
> [10] Kolt, Governing AI Agents (2025).
>
> [11] Geertz, "The Interpretation of Cultures" (1973).
>
> ---
>
> *Update, Nov 27: We have rewritten portions of the abstract to emphasize practical motivations.*
>
> *Update, Nov 29: We have rewritten the introduction to emphasize practical motivations.*
>
> *Update, Dec 2:* We offer a few more highly relevant practical use cases: AI usage in rendering arbitration awards and resolving dispute cases on platforms like Amazon and Ebay.
>
> [12] H Eidenmuller, F. Varesis, (2020)
>
> [13] Westermann, "LLMediator: GPT-4 Assisted Online..." (2023)

---

> ### Author Response · Authors · 2025-11-23
> **Response to Reviewer ck6d: Weakness 1, continued**
>
> **Weakness 1, continued**
>
> > another recommendation: what about the performances of current reasoning models?
>
> We think this a valuable line of inquiry to follow, as indicated in our Discussion section. We wanted to give reasoning models the treatment they deserved, entailing another set of ~15 experiments, which, at the time of submission, was beyond both the scope and budget of our analysis. Nonetheless, time permitting in the review period, we will follow up with some initial experiments that could provide insights for future work on the performance of reasoning models.

---

> ### Author Response · Authors · 2025-11-23
> **Response to Reviewer ck6d: Weakness 2**
>
> **Weakness 2**
>
> > The Values in the Wild is the work from Anthropic researchers using their own tool to analyze the human users and AI. It may only cover what human users perceived and interact with an AI "chatbot assistant", rather than an AI (which could be many roles in the future applications).
>
> We thank the reviewer for raising this important point about the scope of the value taxonomy.
>
> We agree with your points on Values in the Wild, which primarily reflects human-favored assistant values in that setting, rather than a model-agnostic value space for all possible AI roles. We further emphasize that we modified the selection of values specifically for the context of everyday moral dilemmas (that we intentionally subselected). Thus, we do not mean to suggest that these values are a definitive list of values to be used across all cases.
>
> To address this, we:
> - **Explicitly note in the Methods section** that we analyze a subset of assistant-facing, human-elicited values from Values in the Wild, and that our findings are about how models negotiate these values in a chatbot-style deliberation setting.
>
> - Add a brief **limitation in the Discussion** that the taxonomy is built from one model series and user population, and that using a single framework may bias which value patterns are most salient, pointing to future work comparing alternative value taxonomies. We will further note that these values are derived from Claude, and thus may bias how they are inferred from the deliberation outputs.

---

> ### Author Response · Authors · 2025-11-23
> **Response to Reviewer ck6d, Weakness 3**
>
> **Weakness 3**
>
> > Comparison of models experiments is hard to extend to more models due to the bad choice of metric
>
> We agree with the reviewer that it is difficult to compare pairwise change-of-verdict rates. This is why we introduce our multinomial model as a way to test models with a single set of parameters. The multinomial model pools data across all dilemmas, model pairings, and formats, and estimates per-model parameters for inertia (tendency to repeat a previous verdict) and two types of conformity (sensitivity to previous-round verdicts and within-round verdicts). This gives us a global, order-aware comparison of models along deliberative dynamics of interest.
>
> Ultimately, we do not frame our approach in a competitive fashion. We are less interested in which model “wins” the debate, and **more interested in how multi-turn conversation can drive the evolution of blame assignment and values**. A rank-based metric would reduce the complexity of this interaction into a single scalar “strength” or “score,” implicitly assuming a zero-sum winner/loser structure for each debate. In our setting, all models can revise their verdicts. They can move closer to or further from consensus at different times, and models differ along multiple axes (baseline verdict preferences, inertia, and different types of conformity), so collapsing all of this into one rank would obscure the phenomena we are trying to study.
>
> In addition, one of our contributions is that deliberation format matters: synchronous and round-robin interactions induce very different patterns of conformity and inertia. **Elo-style metrics do not naturally account for explicit speaking order effects**, which are crucial in the round-robin setting. They are also **ill-defined in our three-way deliberations**. In three-way debates, for example, a final consensus verdict may be a value compromise that no model proposed in Round 1; models may switch verdicts multiple times; and influence can be shared rather than attributable to a single “winner.” These rich deliberation archetypes are hard to reconcile with a simple arena-style ranking, but are directly captured by the multinomial model through its order-sensitive parameters.
>
> We thank the reviewer for mentioning LitmusValues as highly relevant and valuable related work. We will add this related work to our paper, and add text to the Discussion section. We note that LitmusValues uses such rank-based metrics in a clear binary formulation to rank value preferences of models, in contrast to the proposed use case here which would be to compare “winners” of the deliberation according to their blame assignment.
>
> **Nonetheless, it is useful to include an additional metric for comparison. We aim to calculate Elo ratings in the synchronous setting (where order does not matter), and will update the reviewer once we have added it to the paper.**
>
> ---
>
> *Update, Nov 25:* We have calculated Elo ratings for the three models in the pairwise synchronous settings. We found that GPT had the highest rating (1544), followed by Claude (1517) and lastly Gemini (1438), in agreement with the CoV rates. **We report this result in Section 4.1, third paragraph, and provide additional details in Appendix L.**

---

> ### Author Response · Authors · 2025-11-23
>
> **Weakness 4**
> > followed by W3, authors only compared 3 models. It is insufficient and potentially misleading to compare them as model series difference (e.g. GPT-4.1 directly concludes as GPT, Claude 3.7 Sonnet directly concludes as Claude, and Gemini 2.0 Flash directly concludes as Gemini)
>
> This is fair criticism. Our choice in doing this was not to suggest that our findings were reflective of the entire series. It was simply to save space in both the main text and in the figures (we struggled with this in earlier versions of the figures, finding that abbreviations such as “G2F” for Gemini 2.0 Flash were too confusing). **We will add a sentence clarifying that this is not a reasonable takeaway**. We will also go back through the main text, and strategically refer to the full model name whenever we draw a specific conclusion to ensure the reader does not infer the finding for the model family as a whole.
>
> ---
>
> *Update, Nov 25: We have added a sentence in the results clarifying our reason for the abbreviations.*
>
> *Update, Nov 27: We have rewritten portions of the abstract to account for this concern.*
>
> ---
>
> > several studies [2,3,4] already show that different models in the same model family could have different value preferences.
>
> We agree, as we note this as both an observation and limitation in our Discussion section.
>
> ---
>
> > recommended some stat tests to justify whether the findings can still hold when comparing across three (and ideally even more models).
>
> Can the reviewer clarify which findings? We have run statistical tests on all results showing figure bars, as indicated by our figure captions.

---

> ### Author Response · Authors · 2025-11-23
>
> **Weakness 4, continued**
>
> > similarly, the findings (line 367-368) between different model pairs are interesting but they could have confounding factors (e.g. the opponent effect on the model being tested).
>
> We thank the reviewer for pointing out this potential confound. We agree that the value and verdict patterns we report are naturally conditional on the interaction context, including which opponent a model is paired with and the deliberation format, hence the arrangement of our figure as a comparison (rather than, say, a pure count of value occurrences).
>
> Our goal was not to claim that, for example, “GPT-4.1 always” or “Claude 3.7 Sonnet always” uses a given value profile in isolation, but rather to characterize how each model behaves in interaction with specific counterparts under a given protocol. The multinomial model already captures some of this dependence through dilemma-level fixed effects and format-specific conformity terms, but it does not remove the fact that opponent identity is part of the setting being studied. **We adjusted language in the Results section to emphasize that the reported inter-model differences in values should be interpreted as properties of particular model-opponent pairings and deliberation formats, rather than as unconditional traits of the models in all contexts.**

---

> ### Author Response · Authors · 2025-11-23
>
> **Weakness 4, continued**
>
> > section 4 (steering system prompt) could have more in-depth analysis and experiment. For instance, try different prompt variations. Some prior work [3] has done some explorations on system prompts steerability on value preferences using model spec. Authors can take references of them to provide more in-depth analysis to benefit the community and provide more actionable advice to justify their study's practical importance [W1]
>
> We thank the reviewer for pointing this out, and the suggestion for additional analysis. See our response to Reviewer JE5M for an initial ablation that we have added to the steering section.
>
> We note that our current steering experiments are oriented more toward deliberation dynamics rather than steering toward particular values. We will explore leveraging the techniques in DailyDilemmas but note that a full treatment, to the level provided by that paper, will likely be out of scope for this work.
>
> ---
>
> *Update, Nov 28:* We repost our response to this feedback from Reviewer JE5M for convenience.
>
> We have now conducted a series of steering experiments that we hope address the reviewer's concerns. These include:
> - **An ablation we where omit the goals section of the system prompt: Appendix B, Figure 13.** We found that this ablation largely unchanged change-of-verdict rates.
> - **An experiment where we steered models to be more adversarial to each other: Appendix B, Figure 15 and Table 2.** We found that change-of-verdict rates generally decreased (in the case of GPT-4.1, they went to 0). We found consensus formation significantly decreased in this setting. However, relative differences between models persisted (CoV rates). See below for the no-consensus rates in all settings (higher means models failed to reach a consensus more often).
> | Pairing                           | Original | Balanced | Adversarial |
> |-----------------------------------|----------|----------|-------------|
> | Claude 3.7 Sonnet vs. GPT-4.1     | 0.094    | 0.062    | 0.184       |
> | Claude 3.7 Sonnet vs. Gemini 2.0 Flash | 0.115    | 0.129    | 0.154       |
> | GPT-4.1 vs. Gemini 2.0 Flash      | 0.174    | 0.073    | 0.239       |
> - **An experiment where we steered toward a certain value: Appendix B, Figure 16 and Table 3.** We found that deliberation outcomes were largely unchanged (Figure 16), but that usage of the *Empathy and Understanding* value (the value we steered toward) strongly increased.
> | Model                                    | Original | Steered | % Increase |
> |------------------------------------------|----------|---------|------------|
> | Claude 3.7 Sonnet (vs. GPT-4.1)          | 0.297    | 0.403   | 26.2%      |
> | GPT-4.1 (vs. Claude 3.7 Sonnet)          | 0.262    | 0.492   | 46.7%      |
> | Claude 3.7 Sonnet (vs. Gemini 2.0 Flash) | 0.293    | 0.378   | 22.6%      |
> | Gemini 2.0 Flash (vs. Claude 3.7 Sonnet) | 0.290    | 0.396   | 26.7%      |
> | GPT-4.1 (vs. Gemini 2.0 Flash)           | 0.244    | 0.477   | 48.9%      |
> | Gemini 2.0 Flash (vs. GPT-4.1)           | 0.311   | 0.433   | 28.2%      |
> Together, these results demonstrate that deliberative dynamics and values can both be steered by prompting, though in the case of the dynamics (CoV rates), the differences may not be meaningful across models.

---

### Official Review · Reviewer_hdyG · 2025-10-31

**Soundness:** 3
**Presentation:** 4
**Contribution:** 3
**Rating:** 6
**Confidence:** 4

**Summary:**

The proposed approach leverages debate tactics to determine if deliberative dynamics in multi turn settings impact the socio-technical evaluation of LLMs. In particular, the authors leverage everyday situations from the Reddit AITA community as seed situations. Their findings report how deliberation impacts model behavior.

**Strengths:**

1. The use of the AITA subreddit is extremely well motivated and shrewd, as it is a constantly updated stream of multi-value laden scenarios with lots of potential for disagreement.
2. Leveraging the "Values in the Wild" paper is also a smart choice, given the extensive work done therein.
3. The findings regarding consensus arising from a combination of inertia and conformity are quite interesting and definitely merit future study.

**Weaknesses:**

1. The choice of 3 large scale models is a bit under-motivated. Specifically, why not also try the analyses on a popular open-weight model, such as the Qwen or DeepSeek variety? Seeing how well smaller models, as well as open-weight models, perform in this setting could be quite beneficial for the extensibility of the findings presented. This would also help alleviate the reproducibility concerns that the authors themselves bring up in the Discussion.

**Questions:**

N/A

---

> ### Author Response · Authors · 2025-11-23
> **Response to Reviewer hdyG: Weakness 1**
>
> We thank Reviewer hdyG for their review and highlighting our setup with AITA and use of Values in the Wild as strengths.
>
> **Weakness 1**
> > The choice of 3 large scale models is a bit under-motivated. Specifically, why not also try the analyses on a popular open-weight model, such as the Qwen or DeepSeek variety? Seeing how well smaller models, as well as open-weight models, perform in this setting could be quite beneficial for the extensibility of the findings presented. This would also help alleviate the reproducibility concerns that the authors themselves bring up in the Discussion.
>
> The reviewer raises a good point: we did not allocate enough text fully motivating our choice of models. We used GPT-4.1, Claude 3.7 Sonnet, and Gemini 2.0 Flash since they were, at the time, the most advanced non-reasoning models whose cutoff dates did not match the three corresponding providers (OpenAI, Anthropic, Google). We chose these three providers because they are the most widely deployed assistants whose usage incorporates the everyday advice and moral-guidance settings we study. Thus, their deliberative behavior is directly relevant to the sociotechnical questions we raise. **We have added additional text in the Results section articulating the choice of these models.**
>
> Nonetheless, the Reviewer’s point on examining the performance of open-weight models is very valuable. To address this, we have begun examining how DeepSeek performs in these deliberation settings. Currently, **we have run our head-to-head synchronous experiments between DeepSeek and the above three models**. Our results are expounded in **Section 4.5**, with additional plots detailed in **Appendix D**. We find that DeepSeek is similar to GPT-4.1 in that it relies heavily on the “NTA” verdict (even more so: it uses the verdict roughly 90% of the time in Round 1: Table 4). In the synchronous setting, DeepSeek is more inertial than Claude 3.7 Sonnet and Gemini 2.0 Flash, exhibiting much lower change-of-verdict rates (Figure 14).
>
> This experiment is a first start to addressing the Reviewer’s feedback. We aim to complete the following in the next week:
> - Carry out head-to-head round-robin experiments between DeepSeek and the three current models. In particular, it will be interesting to see whether DeepSeek, like GPT, exhibits strong format-dependent conformity and inertia patterns.
> - Characterize the conformity and inertia of DeepSeek with these results using our model
> - Run similar experiments using a smaller, open-sourced language model in similar head-to-head experiments.
>
> We leave a broader sweep over the full open-model landscape to future work due to cost and time constraints.

---

> > ### Comment · Reviewer_hdyG · 2025-11-26
> >
> > I thank the reviewers for engaging and running additional experiments. I will retain my score accordingly.

---

> ### Author Response · Authors · 2025-11-30
> **Response to Reviewer hdyG: Weakness 1, Update**
>
> **We have updated the paper with a series of experiments with open-source models.** We report them in full.
>
> First, **we conducted synchronous and round-robin deliberations between DeepSeek-V3.2 and the three main models** (GPT-4.1, Gemini 2.0 Flash, and Claude 3.7 Sonnet.
> -  We found that DeepSeek-V3.2 is comparable to GPT-4.1, with very low change-of-verdict rates a similar verdict profile (with a heavy emphasis on the ``NTA'' verdict).
> - In contrast to GPT-4.1, DeepSeek-V3.2 also exhibits low change-of-verdict rates in round-robin deliberation, demonstrating low conformity.
> -  We fit conformity and inertia parameters, finding that DeepSeek is highly inertial ($\alpha_{\text{DeepSeek}} = 2.29$) and lacks conformity ($\gamma_{\text{prev,DeepSeek}} = 0.881$, $\gamma_{\text{within,DeepSeek}} = 0.497$).
> - **Thus, DeepSeek has noticeably different deliberative dynamics relative to the other three models.**
>
> Next, we assessed whether model size had an effect on deliberation. **We ran head-to-head synchronous deliberations between the three main models and both Llama 3.1 8B and Llama 3.1 70B.**
>
> - We found that Llama 3.1 8B failed to reach consensus more often than all other models (28-31%), roughly twice as often as Llama 3.1 70B (8-15%). Llama 3.1 70B's consensus rate is comparable to the other models.
> - At the same time, Llama 3.1 8B had the highest change-of-verdict rate across all models (~45%). Llama 3.1 70B had a lower change-of-verdict rate that was comparable to Gemini 2.0 Flash.
> - These findings appear in tension, and are explained by the fact that Llama 3.1 8B frequently changes its verdict even in deliberations that ultimately fail to reach consensus. This dynamic occurs in 21% of such cases, compared to a baseline of roughly 5–8% for other models (including Llama 3.1 70B).
> - Together, these results provide some evidence that **model size (or ability) may limit the capacity for consensus-making or consistent deliberative dynamics.**
>
> **These results are presented in Section 4.6, with further details in Appendix D and Appendix E.**

---

### Author Response · Authors · 2025-12-03
**General Summary of Review Response**

We thank all the reviewers for their thoughtful feedback. We have worked extensively to address their concerns, and believe the new experiments and analyses have improved the quality of the paper. In this comment, **we summarize the major changes we have made** to address the reviewer concerns.

1. **Deliberations with open-source models.** Reviewer hdyG asked us to consider open-source models. We conducted a range of experiments to address this concern:
   - We ran synchronous and round-robin deliberations between our three main models and **DeepSeek-V3.2**, finding its verdict profile to rely heavily on the NTA label, and exhibits low change-of-verdict rates.
   - We quantified **DeepSeek-V3.2’s inertia and conformity parameters** using these deliberations by extending our multinomial model analysis, finding it to exhibit high inertia and low conformity (in all settings).
   - We assessed how **model size** shapes deliberation outcomes using **Llama 3.1-8B and Llama 3.1-70B**, providing evidence that smaller models may be more volatile and less able to support stable, consensus-forming deliberation dynamics.
2. **Steering experiments.** Reviewers ck6D and JE5M asked for a more robust examination of the steerability of deliberation outcomes via the system prompt.
   - We considered **three different ablations**: one with no goals specified, one steering toward consensus, one steering away from consensus (adversarial). All three were successful, though the relative differences between models remained consistent.
   - We considered **steering toward usage of a certain value** (motivated by Reviewer ck6D), finding that we could significantly increase usage of that value according to the LLM judge.
3. **Validation of LLM judge.** Reviewer N9qX asked for validation and robustness checks on LLM judge for value assessment.
   - We conducted **human annotations** of a subset of samples, finding moderate agreement (on the order of 3-4 values out of 5 shared, roughly).
   - We conducted **robustness checks** with the existing LLM judge (Gemini 2.5 Flash), using a **repeated run** as well as a **run at temperature 0**, finding high agreement (on the order of 4 out of 5 values shared, roughly).
   - We repeated the analysis with a **different judge, GPT-5,** finding moderate agreement on the values (on the order of 3-4 values out of 5 shared, roughly).
   - We further found that the **value alignment analysis was consistent with GPT-5,** suggesting that these findings are robust to the choice of judge.
4. **Evaluation without Deliberation as Baseline.** Reviewer JE5M asked for a baseline established by the models evaluating the dilemmas outside the deliberation context.
   - We conducted **three repeated runs**, for each of the main models, of isolated evaluations of the dilemmas (with a slightly different system  prompt).
   - GPT-4.1 and Claude 3.7 Sonnet exhibited **high agreement with the verdicts** rendered in the first round of synchronous deliberation, suggesting that this is a good placeholder for individual evaluation. Gemini 2.0 Flash exhibited **comparatively lower agreement**, providing some insight into its high change-of-verdict rate.
5. **Robustness of value analysis.** Reviewer JE5M raised concerns about our value similarity metric, given that it does not consider related values, potentially deflating similarity scores.
   - We created a **modified similarity metric** that allowed for increased similarity when values were in the same cluster of the *Values in the Wild* taxonomy.
   - Our results hold under this new metric.
6. **Practical importance.** Reviewers ck6d and N9qX asked us to consider the practical importance of this framework.
   - We have **restructured the Introduction, Related Works**, and added to the **Discussion** section to motivate our approach by grounding further in practical use cases and deployed agentic systems that incorporate the formats we test in this work.

Once again, we thank the AC and reviewers for their hard work throughout this entire process!

---

### Meta-Review · Area_Chair_DLLQ · 2025-12-29

**Summary:**

The decision to reject is primarily informed by the fact that the study's empirical scope and methodological rigor are insufficient to support its sociotechnical claims, specifically the reliance on an unvalidated LLM judge for value classification which introduces potential bias. Reviewers expressed significant concern regarding the lack of practical utility in the experimental design, noting that the metrics focused on inertia and conformity without evaluating user-centric outcomes, and criticized the pairwise comparison metrics as difficult to interpret compared to standard rank-based alternatives.

**Reviewer Concerns:**

The rebuttal effectively addressed specific empirical gaps by incorporating open-weights models. However, the critical concern regarding the practical significance of the findings remains outstanding, as the study still lacks a demonstration of how the observed deliberation dynamics translate into tangible improvements in alignment or user welfare, and the fundamental reliance on a specific, potentially biased value taxonomy derived from a single model family continues to limit the broader validity of the conclusions.

**Reviewer Scores:**

Reviewer hdyG would maintain their score as they explicitly stated they would retain it despite the additional open-source experiments;
Reviewer ck6d would likely maintain their score because the addition of Elo ratings does not fully resolve the deeper skepticism regarding the practical necessity and generalizability of the debate formats;
Reviewer JE5M would likely maintain their score as the concerns regarding the inherent biases of the Reddit dataset and the simplistic value similarity metrics were not fundamentally rectified by the modified analysis;
Reviewer N9qX would probable lower their score, as the additional annotation experiment yielded negative results regarding empathy improvement, directly confirming the critique that the proposed mechanism fails to generate tangible user-centric welfare gains.

---

### Decision · Program_Chairs · 2026-01-26

Reject